# Evaluating the reliability of U-Pb LA-ICP-MS carbonate geochronology: matrix issues and a potential calcite validation reference material.

Marcel Guillong[1], Jörn-Frederik Wotzlaw[1], Nathan Looser[1], Oscar Laurent[1]

[1]Department of Earth Sciences, ETH, Zürich, 8092, Switzerland

*Correspondence to*: Marcel Guillong (guillong@erdw.ethz.ch)

**Abstract**

We document that the reliability of carbonate U-Pb dating by laser ablation inductively coupled plasma mass spectrometry (LA-ICP-MS) is improved by matching the aspect ratio of the LA single hole drilling craters and propagating of long-term
excess variance and systematic uncertainties. We investigated the impact of different matrices and ablation crater geometries using U-Pb isotope analyses of one primary (WC-1) and two secondary reference materials (RMs). Validation RMs (VRMs) include a previously characterized (ASH-15D) and a new candidate (JT), characterized by ID-TIMS (intercept age: 13.797 ± 0.031 Ma) with excellent agreement to pooled LA-ICP-MS measurements (13.81 ± 0.11 ¦ 0.30 Ma), U concentration of approx. 1 µg/g and $^{238}U/^{206}Pb$ ratios from 5 to 460, well defining the isochron. Differences in ablation crater depth to diameter ratios
(aspect ratio) introduce an offset due to downhole fractionation and/or matrix effects. This effect can be observed either when the crater size between U-Pb RM and sample changes or when the ablation rate for the sample is different than for the RM. Observed deviations are up to 20% of the final intercept age depending on the degree of crater geometry mismatch. The long-term excess uncertainty was calculated to be in the range of 2% (ASH-15D) to 2.5% (JT), and we recommend propagating this uncertainty into the uncertainty of the final results. Additionally, a systematic offset to the ID-TIMS age of 2-3% was observed
for ASH-15D but not for JT. This offset might be due to different ablation rates of ASH-15D compared to the primary RM or remaining matrix effects, even when chosen aspect ratios are similar.

## 1 Introduction

Recent improvements in the sensitivity of ICP-MS instruments coupled to a laser ablation system allows not only to date very young zircons by the U-Pb method (Guillong et al., 2014), but also minerals with very low U concentrations, typically of
several ppb to tens of ppm, and even lower concentrations of radiogenic Pb, such as carbonates (Li et al., 2014;Methner et al., 2016;Roberts and Walker, 2016;Nuriel et al., 2017). Additionally, carbonates often incorporate variable amounts of initial (common) Pb during crystallisation from aqueous fluids, so that age determination relies on the use of regression lines (isochrons) in the Tera-Wasserburg isotopic space ($^{207}Pb/^{206}Pb$ vs. $^{238}U/^{206}Pb$) (Li et al., 2014). Importantly, this also entails that there is no available carbonate reference material (RM) yielding a concordant U-Pb age. Therefore, accurate U-Pb dating

of carbonates requires a two-step data reduction approach (Roberts et al., 2017) consisting of (1) $^{207}Pb/^{206}Pb$ mass bias correction based on a homogeneous reference material (typically a standard glass); and (2) U/Pb inter-element fractionation correction based on the lower intercept in the Tera-Wasserburg concordia diagram using a matrix-matched RM. With this method, carbonates can be dated by LA-ICP-MS with the advantage of easy availability, high sample throughput, high spatial resolution allowing to resolve large differences in U-Pb ratio and cost effectiveness. Within a day, 300-600 single points can be analysed and a minimum number of 20-30 points per sample is suggested (Beaudoin et al., 2018;Godeau et al., 2018;Yokoyama et al., 2018), although the appropriate number is strongly sample-specific and depends on the variability of the initial to radiogenic Pb ratios. To maximize the variation of the initial to radiogenic Pb ratios and improve the isochron, new approaches in the acquisition of data by imaging and data pooling were introduced and look promising (Drost et al., 2018). Previous studies of dating carbonates, mostly calcite, focused either directly on the analytical method development (Li et al., 2014;Roberts et al., 2017;Drost et al., 2018;Yokoyama et al., 2018) or a range of applications from dating fault activity, constraining the timing of hydrothermal mineralisation to directly dating early-diagenetic cements in ammonites (Coogan et al., 2016;Methner et al., 2016;Burisch et al., 2017;Drake et al., 2017;Goodfellow et al., 2017;Nuriel et al., 2017;Hellwig et al., 2018;Walter et al., 2018;MacDonald et al., 2019;Scardia et al., 2019;Burisch et al., 2018). Only recently, validation RMs (VRMs) were routinely analysed e.g. (Beaudoin et al., 2018) with data reporting standards following the community derived standards suggested for zircons (Horstwood et al., 2016). However, no actual data on the long term excess variance (ε`) is given as not many VRMs are available and as the number of sessions including these was limited (Beaudoin et al., 2018). In this work, we introduce a new VRM and aim at investigating the long-term excess variance of U-Pb LA-ICP-MS carbonate dates. In addition, we investigate potential matrix effects influencing the accuracy of such dates, which have been largely overlooked so far. In particular, we document that changes in laser crater aspect ratios between primary RMs and samples may result in significant inaccuracy of U-Pb LA-ICP-MS carbonate dates and propose new analytical strategies to minimize this effect.

## 1 Instrumentation, methods and samples

### 1.1 LA-ICP-MS analyses

The LA-ICP-MS analyses were carried out at ETH Zürich, using a Resolution laser ablation system with a 193 nm excimer (ArF) laser source and a two-volume, Laurin Technic S-155 ablation cell. To investigate possible matrix effects and the influence of the ablation crater geometry, we used variable crater sizes and repetition rates of the laser producing laser ablation craters with aspect ratios (crater depth / crater diameter) ranging from 0.08 to 1.3. Crater geometries (depth and diameter) were measured using a Keyence Digital Microscope VHX-6000.

The ablated aerosol was mixed in the ablation cell with carrier gas consisting of helium (0.5 L.min$^{-1}$) and make-up gas consisting of argon (ca. 1 L.min$^{-1}$) and nitrogen (2 mL min$^{-1}$). The aerosol was then homogenized by a squid smoothing device and introduced in the plasma of the Thermo Element XR ICP-MS. This single collector sector field MS is equipped with a

high capacity (80 m$^3$ h$^{-1}$) interface pump to achieve, in combination with jet sampler and normal H-skimmer cones, a detection efficiency in the range of 2% (based on U in a single spot ablation of NIST SRM612 glass). Detailed instrumentation and data acquisition parameters are summarized in Table A1. A measurement session consists of several samples (20-50 points analyses each), matrix-matched primary RM (WC-1; n = 20-40), two validation RMs (ASH-15D (Mason et al., 2013;Vaks et al., 2013) and JT; n=15-30) and a homogenous glass RM (NIST614 or NIST 612; n = 10-20). RMs are distributed over the whole sequence to monitor and if necessary, to correct for instrumental drift.

## 1.2 LA-ICP-MS data reduction

The data reduction methods follow largely the one described in Roberts et al. (Roberts et al., 2017) and are described here shortly focusing on differences and some observations. The raw data (time-resolved intensities) from the spectrometer are loaded into the Iolite v.2.5 data reduction software (Paton et al., 2011) and processed using the VisualAge data reduction scheme (Petrus and Kamber, 2012;Paton et al., 2010), to calculate only gas blank corrected intensities, raw ratios, and their uncertainties. The integration intervals of both reference materials and samples are subsequently adjusted to optimize the spread along the isochron line. The selection of different length or depths for integration intervals for primary RM and the sample, or splitting a single spot ablation into several separate intervals, can introduce systematic offsets due to different downhole fractionation between the primary RM and the sample. Although best practice would be to use integration intervals that are as close as possible identical with respect to crater shape for both the RM and the sample, the potentially introduced offsets would average out between the different spots for a given sample due to the virtually random distribution of favourable signal intervals (high U, low initial Pb).

Further data reduction including drift and matrix correction uses an in-house spreadsheet based on Microsoft Excel fallowing the protocols in (Roberts et al., 2017). The $^{238}$U/$^{206}$Pb ratio is drift corrected using the glass RM, e.g. NIST 614. Due to the low $^{207}$Pb count rate on some of the samples and RMs, the $^{207}$Pb/$^{206}$Pb ratio is calculated as ratio of the mean count rate for each integration interval and not the mean of the ratios of each sweep. The $^{207}$Pb/$^{206}$Pb ratio is corrected for mass bias using the mass bias factor determined from the homogenous glass RM. The drift-corrected $^{238}$U/$^{206}$Pb ratios and the mass bias-corrected $^{207}$Pb/$^{206}$Pb ratios of all analyses of the WC-1 calcite RM are subsequently plotted in a Tera-Wasserburg concordia diagram, using IsoplotR (Vermeesch, 2018). An isochron is calculated by linear regression through the resulting dataset and anchoring to the initial $^{207}$Pb/$^{206}$Pb of 0.85 ± 0.04 (Roberts et al., 2017). The ratio between the lower intercept age obtained thereby and the reference age of 254.4 ± 6.4 Ma (Roberts et al., 2017) is used as the correction factor for the $^{238}$U/$^{206}$Pb ratio throughout the sequence. This correction factor would encompass matrix effects, including laser-induced (i.e. downhole) element fractionation, and ICP-related U/Pb inter-element fractionation, including mass bias.

## 1.3 ID-TIMS

For the characterization of JT as a new potential VRM we used isotope dilution thermal ionization mass spectrometry methods equivalent to those described in (Nuriel et al., in prep). Seven calcite chips of 1.3 to 3.7 mg were sampled from the JT calcite

vein using a stainless-steel needle. Individual chips were loaded in 3 ml Savillex beakers and repeatedly cleaned in ultrapure

acetone and water. Cleaned aliquots were spiked with 4-8 mg of the EARTHTIME $^{202}Pb$-$^{205}Pb$-$^{233}U$-$^{235}U$ tracer solution and dissolved in 6N HCl at 120°C for 30 minutes. Dissolved samples were dried down and re-dissolved in 1N HBr for anion exchange chromatography. U and Pb were separated using a HBr-HCl based anion exchange chemistry employing AG1-X8 resin in 50 µl Teflon columns. The U and Pb fractions were collected separately in 7 ml Savillex beakers and dried down with a drop of 0.02 M $H_3PO_4$. U and Pb were loaded on outgassed Re filaments with a Si-Gel ion emitter for thermal ionization

mass spectrometry and analyses were performed using a Thermo TRITON plus instrument at ETH Zürich. Pb isotope ratios were measured on the axial secondary electron multiplier and U was measured as $UO_2$ employing a static routine with Faraday cups connected to amplifiers with $10^{13}$ ohm resistors(Wotzlaw et al., 2017). Data reduction and uncertainty propagation was done using Tripoli, an excel-based spreadsheet that employs algorithms of (Schmitz and Schoene, 2007) and isochron calculations were performed using IsoplotR (Vermeesch, 2018). U-Pb data were not corrected for $^{234}U$ and $^{230}Th$ disequilibria.

Uncertainties are reported at the 95% confidence level without systematic uncertainties associated with tracer calibration and decay constants unless otherwise indicated.

### 1.4 New validation reference material JT

JT is a vein calcite that originates from a deep borehole in the northern Swiss Molasse Basin and is hosted by a micritic limestone of the Middle Triassic Muschelkalk Group. JT is part of a dense network of calcite veins associated with a thrust

fault branching off from the basal décollement of the Jura fold-and-thrust belt. (Looser et al., in prep).

### 2 Results and discussion

All LA-ICP-MS and ID-TIMS data can be found in the supplementary information Table S1(LA-ICP-MS) and S2 (ID-TIMS).

### 2.1 Characterisation of JT as a validation reference material

We characterized the vein calcite JT both by ID-TIMS and LA-ICPMS (all data provided in the supplementary file). The ID-

TIMS analyses yielded an isochron, lower intercept age of 13.797 ± 0.031 Ma (n=6, 1 outlier excluded; MSWD=3.6), and a $(^{207}Pb/^{206}Pb)_0$ of 0.8394 ± 0.0025 (Fig. 1a). The intercept age of pooled LA-ICP-MS data from 16 sequences with a total of 474 single point analyses (13.75 ± 0.11 Ma respectively 13.75 ± 0.36 Ma including excess variance ε`, see below; MSWD = 2.0; $(^{207}Pb/^{206}Pb)_0 = 0.8473 \pm 0.008$; Fig. 1b) is identical within uncertainty to the mean of the intercept ages from the same 16 sequences (13.695 ± 0.157 Ma, respectively 13.70 ± 0.37 Ma including ε`, Fig. 2a). Both well overlap with the ID-TIMS

intercept age. Additional measurements of 10 spots on 43 pieces of JT available for distribution shows that some pieces are dominated by initial Pb and the Overall, the U concentration ranges from below 0.01 µg/g up to 5 µg/g with a mean of 0.6 ppm and a median of 0.44 µg/g. The $^{238}U/^{206}Pb$ ratio ratio varies between 0.04 and 455. Detailed description of the results (Table S3) also contain isochrons for each individual pieces can be found in the supplementary material.

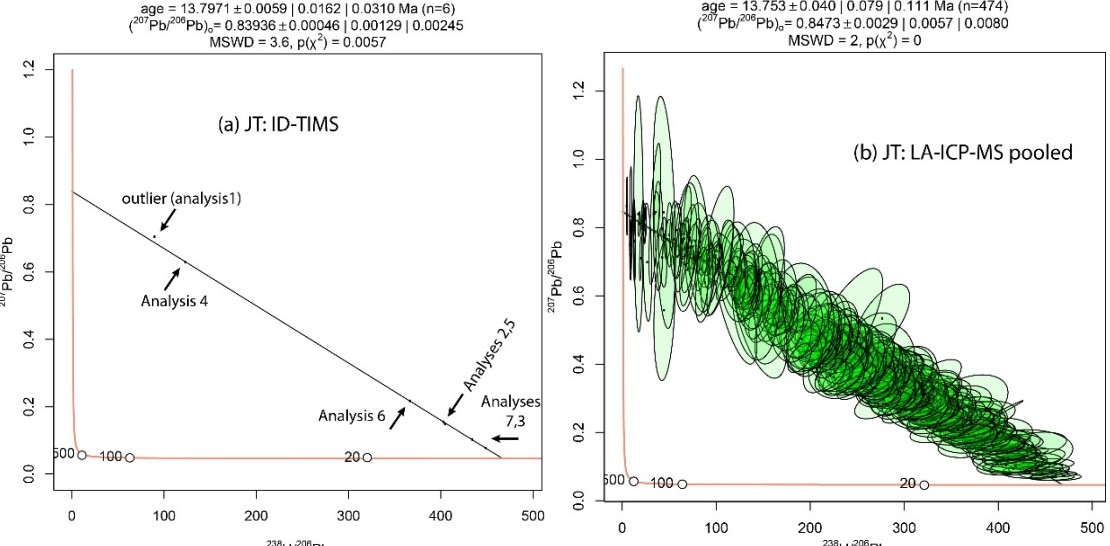

Figure 1: Tera-Wasserburg concordia plot of JT analyses by (a) ID-TIMS (n=6) and (b) LA-ICP-MS (n=474).

## 2.2 Long-term excess variance

The VRMs JT and ASH-15D were analysed 16 and 30 times in 9 and 16 sessions respectively, sometimes with different laser parameters, each yielding an intercept age with uncertainty. Fig. 2 shows the sorted intercept ages with uncertainties both without (white) propagation of the long-term (inter sessions) excess variance ($\varepsilon$`) and with propagation (green). The MSWD of the original data sets are 1.57 for JT and 1.26 for ASH-15D. Assuming ideal behaviour of these VRMs (homogeneous age, cogenetic character of all analysed domains and closed system behaviour) a MSWD of ~1 is expected. To obtain an MSWD of ~1, a long term (inter session) excess variance ($\varepsilon$`) of 2.5 % (JT) and 2.0 % (ASH-15D) needs to be propagated by quadratic addition to the intercept uncertainty of the individual sequences (Fig. 2). For this calculation, only VRM measurements with similar aspect ratio as the primary RM were considered. This excess variance is slightly larger than for zircons of ca. 2% 2S (Horstwood et al., 2016), probably reflecting the difference in age calculation, heterogeneity of RM WC-1 (See section 2.5), heterogeneity of VRMs and matrix differences. The latter two potentially include the selection of integration intervals that do not systematically match those of the primary RM, resulting in slight offsets (see section 1.2).

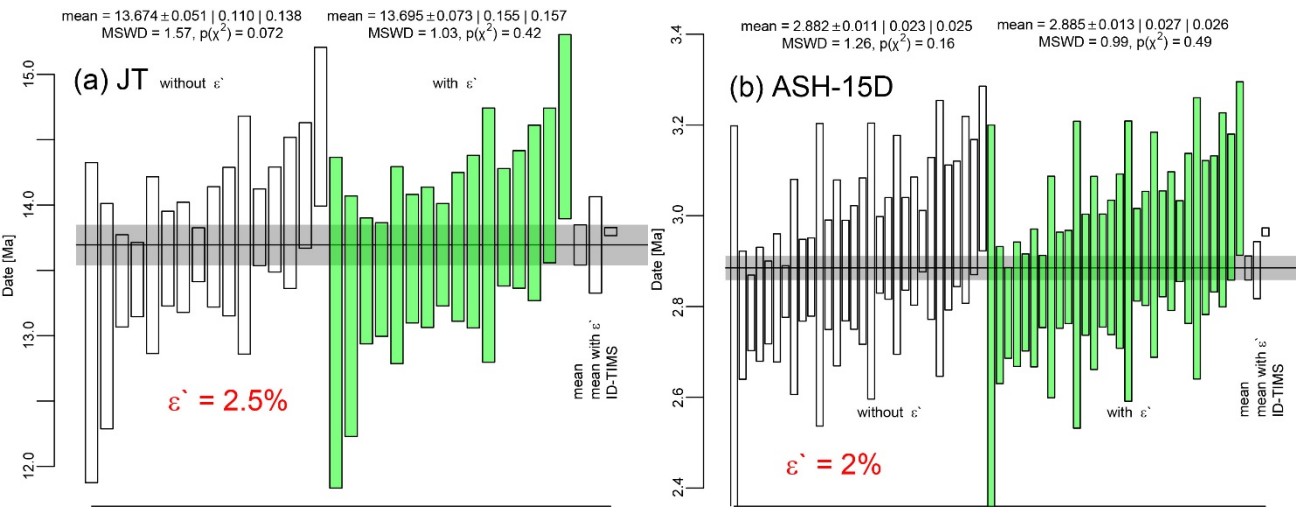


**Figure 2: Sorted intercept ages for JT (a: white) of n=16 with an MSWD of 1.57 and ASH-15D (b: white) of n=30 with an MSWD of 1.26 for the estimation of the long-term excess variance (ε`) to be propagated and the same data including ε` (green). Also presented is the superpopulation mean, mean with ε` and reference age from ID-TIMS. Uncertainties are 95% confidence levels with overdispersion (white) respectively 95% confidence levels with overdispersion and excess long-term variance (green).**

## 2.3 Influence of the ablation crater aspect ratio on data accuracy

### 2.3.1 Results on tests based on WC-1, JT and ASH-15D

During 15 sequences of carbonate dating, we have measured the validation RM with different crater sizes and repetition rates, matching the need of sufficient signal on the samples with generally low U contents. We varied the crater diameter and repetition rate from 110 μm and 5 Hz for the primary RM WC-1 up to 250 μm and 10 Hz for VRMs and samples (exact

parameters are listed for all sessions in Table S1). We first observed that analyses performed with ablation craters diameters larger than those of the primary RM show a systematic offset towards higher $^{238}$U/$^{206}$Pb ratios, corresponding to younger isochron intercept ages. This suggests an influence of the crater geometry on the offset between measured and real values, which is problematic when the unknowns and primary RM are measured with different crater sizes and/or repetition rates Therefore, we further measured a single sequence (190130) analysing WC-1 not only as primary RM with the standard

parameters of 110 μm ablation crater diameter and 5 Hz repetition rate, but also with 163 μm, 3 Hz; 110 μm, 10 Hz, 74 μm, 10 Hz and 51 μm 15 Hz, to systematically investigate the influence of ablation crater aspect ratio. These parameters result in extreme difference of crater aspect ratios (crater depth / crater diameter) between ~0.08 (flat crater, 163 μm, 3 Hz) and ~1.3 (deep crater, 51 μm, 15 Hz). The U-Pb isotopic data, corrected only for $^{207}$Pb/$^{206}$Pb mass bias, yield intercept ages between 226 Ma (flat crater) and 267 Ma (deep crater) as shown in Fig. 3 and Fig. A1. Plotting the aspect ratio mismatch (armm, aspect

ratio of the sample / aspect ratio of the primary RM) vs. the age offset relative to the accepted age, a linear correlation is given not only for WC-1 (Fig. 4a) but also for the validation RM JT (Fig. 4b) and ASH-15D (Fig. 4c).

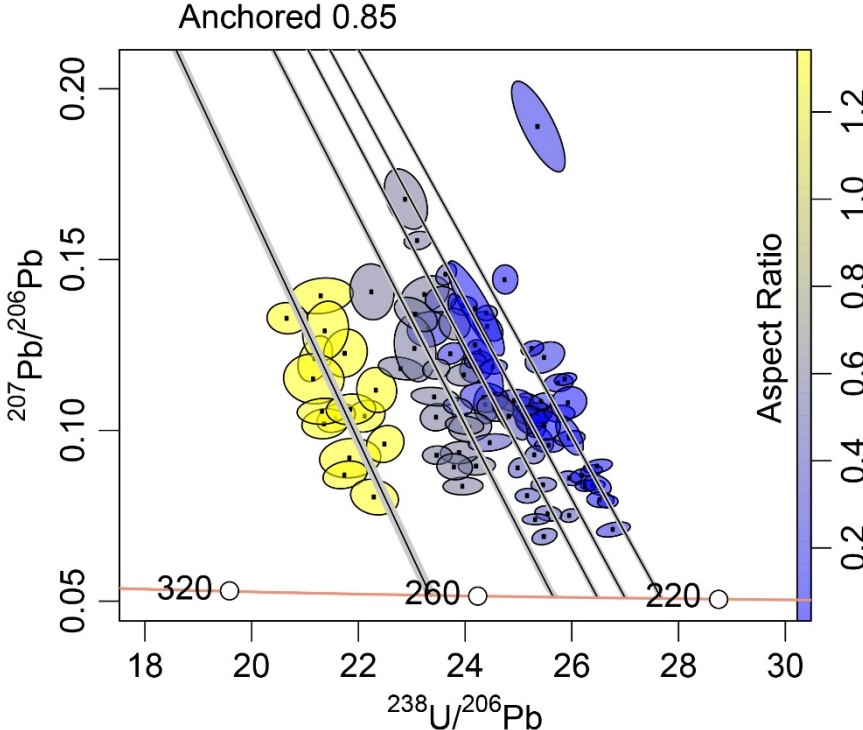

**Figure 3: Tera-Wasserburg concordia plot of WC-1 analyses, corrected only for ²⁰⁷Pb/²⁰⁶Pb mass bias, ablated with different ablation crater aspect ratios. The deeper the crater the older the age. Intercept ages are: ~ 270, 246, 239, 234, 228 Ma. Ellipses are 95% confidence level. Individual plots including regression and intercept statistics are shown in Fig. A1.**

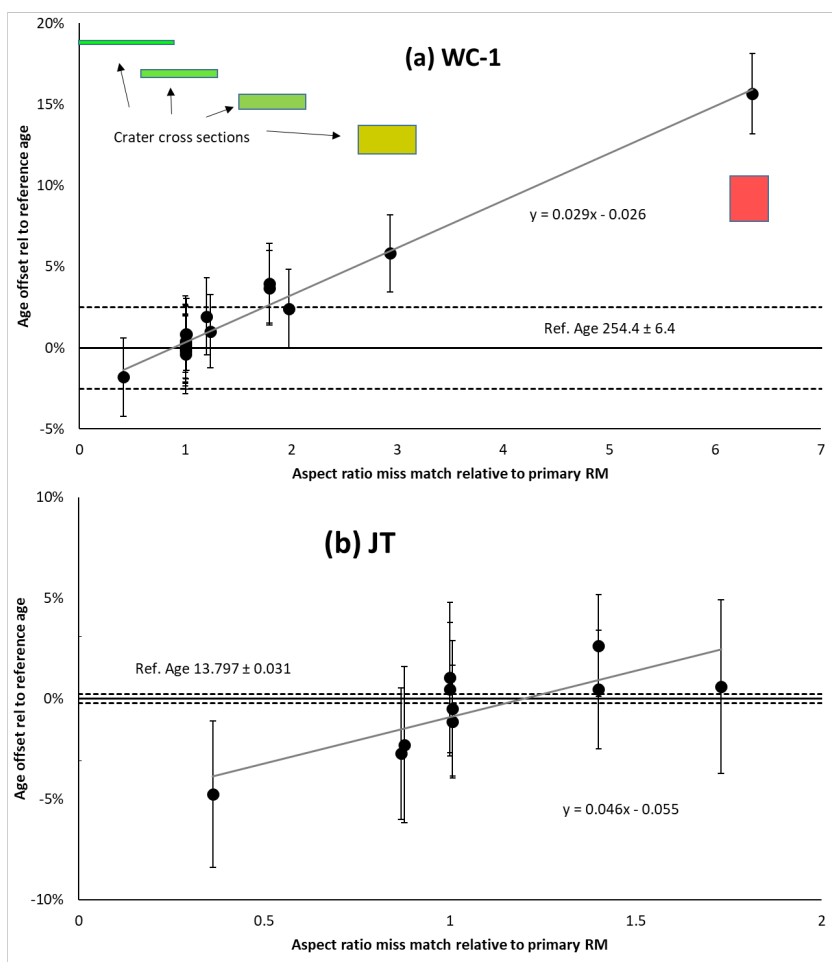

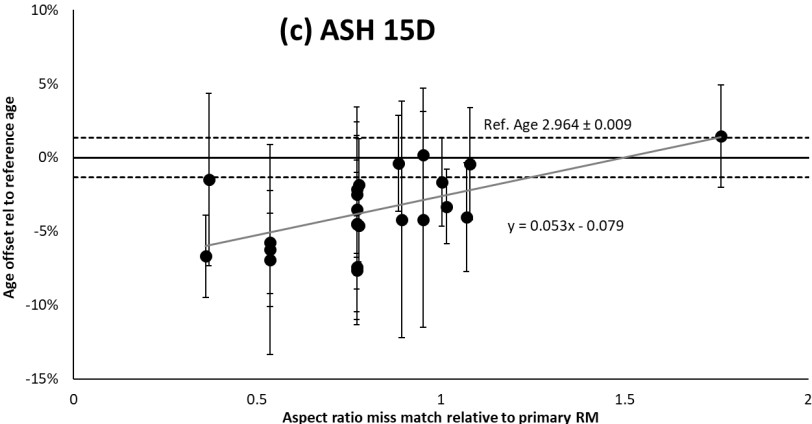

**Figure 4: Plots of relative age offsets relative to the accepted age for various calcite RMs as a function of the aspect ratio offset relative to the primar RM. (a): WC-1 where the coloured rectangles represent the cross section of the crater (not to scale). (b): JT validation RM including the ID-TIMS reference age (this study). (c) ASH-15D validation RM including the ID-TIMS reference (Nuriel et al., in prep). Age error bars are 95% confidence including systematic uncertainties as propagated here (see text for details).**

### 2.3.2 Variable ablation efficiency for different carbonates

Due to the observed age offset depending on the LA crater geometry (Fig. 4), the ablation rate between RM and unknown sample must be very similar when using identical laser parameters (energy density, crater size and repetition rate) to get the same ablation crater aspect ratio. As the ablation efficiency of an unknown sample is not known prior to ablation, we measured ablation rates of various carbonates with differences in mineralogy (calcite, dolomite, aragonite), crystal size (micritic, sparry), and purity (turbid, clear), relative to the ablation rate of WC-1 (ablation rate of ~120 nm per laser pulse at energy density of 2

J cm$^{-2}$) as shown in figure 5, to estimate the ablation rate of a "unknown" carbonate. Regardless of their occurrence (i.e. sparry, micritic, etc.), calcite matrices show ablation rates that vary within <14% relative to WC-1. In turn, the corresponding armm if WC-1 is used as primary RM would result in an offset of the lower intercept date of 1% or less (Fig. 4), which is identical or smaller to the minimum uncertainty on such dates based on our assessment of the excess variance. By contrast, dolomite shows higher ablation efficiencies than calcite and much larger variations, from 105 to 160% relative to WC-1 (Fig. 5). The

only aragonite sample tested in this compilation had an even faster ablation rate than dolomite and almost twice as fast as WC-1 (Fig. 5). We presume that these differences in ablation rates could result in significant age offsets in a roughly estimated range of 4-8 % for dolomite (160 % ablation rate compared to WC-1)and 6-11 % for aragonite (200 % ablation rate of WC-1) based on the offsets found in figure 4. However, this hypothesis needs validation and any attempt to date dolomite or aragonite needs careful validation. In addition, the large spread for dolomite and the small number of tested aragonite samples makes

the estimation of the ablation rate unpredictable for these matrices. These results highlight that matrix-matched standardization is required for accurate LA-ICPMS U-Pb geochronology of different carbonates and dolomite/aragonite dates obtained using WC-1 calcite as primary RM might be prone to systematic inaccuracy.

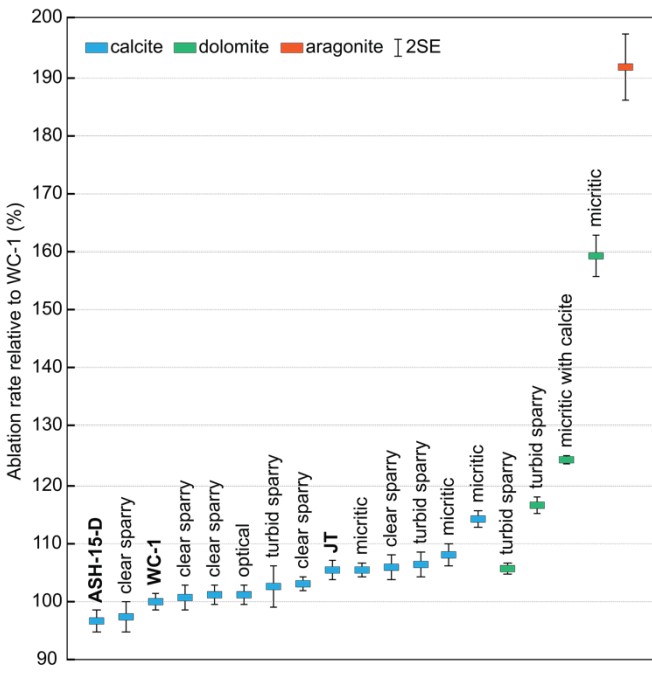

**Figure 5: Ablation rate variation relative to WC-1 for different calcite, dolomite samples and one aragonite sample.**

### 2.3.3 Strategies for matching aspect ratios

The results showed in Figure 4 suggest that carbonate U-Pb dating by LA-ICP-MS is only accurate when the laser ablation crater aspect ratio is similar between RM and unknown sample. If the crater geometry varies, an artificial and significant age offset can be introduced (>10% relative in extreme cases; Fig. 4). This can be a problem because U contents in carbonates vary by orders of magnitude, while only one characterized RM (WC-1) with relatively homogenous and high U content ($3.7 \pm 1$ μg/g, 2S.D.; (Roberts et al., 2017)) is available so far. Therefore, it is not ideal to use one single crater size and repetition rate for all unknowns and RMs. We suggest therefore a two-step strategy to account for this issue: (1) estimating the U content of all unknowns to be dated in a specific sequence, via a short, pre-sequence test run; and (2) apply a crater diameter to each unknown, inversely proportional to its U content, and adjust the repetition rate as to match the aspect ratio of the craters in the primary RM. Based on our experience, the test run needs not be longer than 0.5 h, with only few (3-5) spots per sample using a shortened ablation time (10-15s). Based on this suggestion, it is possible to use larger spot sizes for lower-U samples than used for the RM to improve the counting and precision, as long as the repetition rate is increased accordingly.

It is possible to measure the crater depth and aspect ratios of the analysed samples and RMs post ablation and if necessary, apply a correction (based on the linear relationship between aspect ratio mismatch and age offset as shown in Fig. 4) to improve accuracy, especially when dealing with different carbonate minerals. However, this correction will have to be based on measurements with different aspect ratios for both primary and validation RMs ideally within the same session, as it seems likely that both the ablation rate and age offsets are dependent on the actual laser ablation and ICP-MS parameters. A detailed study on how to best apply this correction if necessary is beyond the scope of this work, and we only suggest for a simpler and more robust data reduction to always use similar aspect ratios. As shown in section 2.3.2 and Figure 5, for various types of calcite the differences in ablation rate relative to WC-1 are sufficiently small that this procedure would lead to accurate ages (i.e. offset by less than 1% relative).

### 2.4 $^{207}Pb/^{206}Pb$ correction

NIST 614 has been proposed as RM for the mass bias correction of the $^{207}Pb/^{206}Pb$ ratio (Roberts et al., 2017) due to the similar Pb concentration to many carbonates. We investigated several homogeneous and well characterized glass RMs (NIST 610, NIST 612, NIST 614, GSD-1G and KL2G) on two different days to be used for this correction and we found no significant influence from different LA parameters, notably ablation crater diameter, and Pb concentration of the RM (Fig.6). Only the smallest spot size of 13 microns on the highest Pb concentration glass (NIST 610) produced a small offset (within uncertainty) compared to the other glasses.

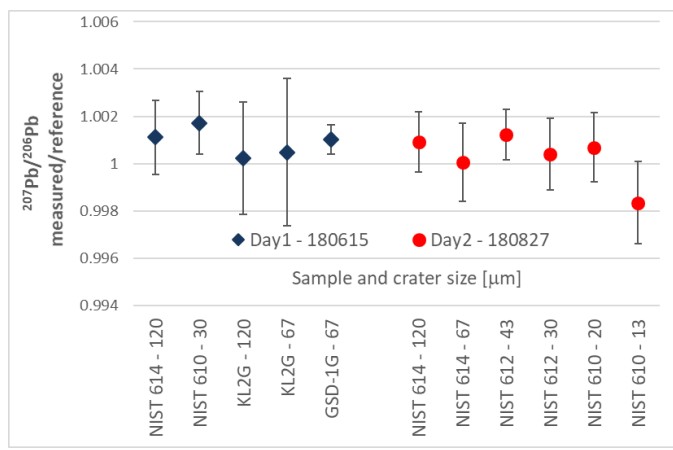


**Figure 6: Plot of the ratio between measured and reference <sup>207</sup>Pb/<sup>206</sup>Pb ratio in different RM glasses and using different crater sizes. The correction factor shows no variation against RM type or crater size. Error bars are 2 SE.**

## 2.5 WC-1 heterogeneity

We document heterogeneity in the currently most used calcite RM WC-1(Roberts et al., 2017). This information is meant as a

caution to analysts to make sure they understand that carbonates are quite heterogeneous materials and are prone to diagenetic alteration including open-system processes at various stages in their geological history resulting in zones with the potential for different ages being recorded. Care needs to be taken as some WC-1 aliquots may have larger age variation than initially described (Roberts et al., 2017).

To investigate the homogeneity of our aliquot of WC-1, in one sequence (190405) we specifically targeted a brighter, vein-

like, more sparry zone, in addition to the normal sampling strategy aiming at the darker, more homogeneous material as shown in figure 7. A white alteration vein cross-cutting through the darker zone with similar appearance like the white zone in our aliquot in the work initially characterizing WC-1 (Roberts et al., 2017) was shown to be high in Th and the transition metals. While for that case, no difference in age is documented, our data shows that the brighter zones may yield significantly younger ages of 203±7 while the darker zones used for matrix correction in this session yielded the expected results of 254.3±2.2,

similar to other sessions and to the results in (Roberts et al., 2017). Given the textural difference between the white and the dark calcites, reflecting different ambient conditions during precipitation, the white zone rather reflects precipitation at different times than Pb loss due to open system behaviour.The analyses of the white zone show a larger scatter than in the dark zones, possibly due to mixing between the two phases in deeper parts of the ablation crater.. This finding further demonstrates that WC-1 as already described (Roberts et al., 2017) is "not the perfect material because of its modest heterogeneity", not

only in chemical composition but also in age.

According to our findings, when using WC-1 as RM the locations for analysis have to be chosen carefully and the data has to be screened for outliers to avoid additional scatter and a bias towards younger ages of the RM, finally resulting in older ages for the unknown samples. Especially points with an increased <sup>207</sup>Pb/<sup>206</sup>Pb ratio that fall off from the isochron are potentially

biased and should be treated with caution. More generally, this shows that additional and more homogeneous RMs are urgently needed for carbonate dating.

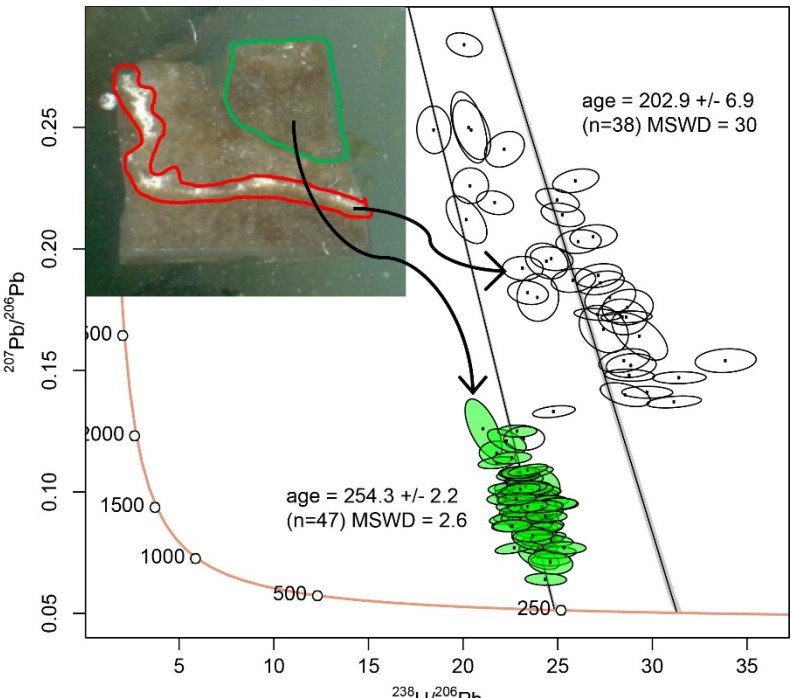

**Figure 7: Tera-Wasserburg concordia plot of WC-1 analyses in two different regions of the aliquot fragment as indicated in the inserted image. The darker part (green ellipses) gives results in agreement with the recommended age (254.4±6.4) while the brighter vein like part (red) gives more scatter and younger ages.**

## 3 Summary and conclusion

We report LA-ICP-MS U-Pb data for previously and newly characterised calcite reference materials. We introduce the JT vein calcite as a potential validation RM due to its homogeneity and very good spread in relative radiogenic to initial Pb contents. The ID-TIMS U-Pb data for JT yielded an isochron intercept date of 13.797 ± 0.031 Ma, consistent with a LA-ICP-MS isochron intercept date of 13.75 ± 0.36 Ma (relative to WC-1).

Repeated LA-ICP-MS analyses of this new VRM and the existing calcite RM ASH-15D over an extended time period show that excess variance of 2 - 2.5% should be propagated in the obtained, individual lower intercept dates. This estimate of the long term excess variance is larger than for other LA-ICP-MS geochronological methods such as U-Pb in zircon (Horstwood et al., 2016) and may encompass a greater heterogeneity of samples, primary RM (WC-1) and different ablation rates between both, as highlighted here.

We also document that a mismatch in the ablation crater aspect ratio between the primary RM and unknowns result in significant age offsets. In theory, this offset can be minimized by by using the same diameter and repetition rate for standards

and unknowns. However, this may not always be possible, especially if the U content is significantly different in unknowns compared to the primary RM, in which case matching the ablation crater aspect ratio of the primary RM is the easiest, more efficient way to get accurate results. For instance, in the case of low U samples, the aspect ratio can be matched by increasing proportionally the laser repetition rate and  crater diameter with the other benefit to yield more signal and thereby decrease analytical uncertainties.. While the correction of the U/Pb ratio is very sensitive to laser ablation parameters, the $^{207}Pb/^{206}Pb$ correction is very insensitive and it is possible to use almost any well-characterized material with well characterized Pb isotopic composition.

The offset of 2-3% (usually within the uncertainty) of LA-ICP-MS results for ASH-15D compared to ID-TIMS results remains, while the results for JT are in very good agreement. This offset cannot be explained completely by differences in ablation rate and may be an additional matrix effect to be investigated in detail in future work.

Furthermore, we show differences in ablation rate between different types of calcites (<14 % relative to WC-1) and, overall, different carbonate minerals (up to + 60% for dolomite and up to + 90% for aragonite, relative to WC-1). While the expected age offset on calcite is negligible, this likely introduces additional systematic errors that needs validation when using non-matrix matched standardization, i.e. using WC-1 calcite as primary RM to date dolomite or aragonite. The latter observation calls for characterization of further carbonate standards including various calcites, dolomites and aragonites. This will significantly increase the accuracy and possible applications of carbonate U-Pb geochronology by LA-ICPMS.

## 4 Data availability

All data used in this manuscript is available from the supplementary files.

## 5 Sample availability

Sample JT is available on request in limited quantities from the author: guillong@erdw.ethz.ch.

## 6 Appendices

Table A1 : LA-ICP-MS parameters

LA-ICP-MS U-(Th-)Pb Metadata

| Laboratory and Sample Preparation | |
| --- | --- |
| Laboratory name | Dept of Earth Sciences, ETH Zurich |
| Sample type/mineral | Carbonates, mostly calcite |
| Sample preparation | thin sections or chips mounted in epoxy |
| Imaging | CL8200 Mk5-2 Optical Cathodoluminescence System |

| Laser ablation system | |
| --- | --- |
| Make, Model and type | ASI (Resonetics) Resolution S155 |
| Ablation cell and volume | Laurin Technic, 2 volume cell, effective volume $ca$. 1 cm$^3$ |
| Laser wavelength (nm) | 193 nm |
| Pulse width (ns) | 25 ns |
| Fluence (J cm$^{-2}$) | ~1.8 J cm$^{-2}$ |
| Repetition rate (Hz) | variable, see data for actual value |
| Ablation duration (s) | 40 s |
| Ablation pit depth / ablation rate | Variable, equivalent to 0.09-0.15 $\mu$m/pulse |
| Spot diameter ($\mu$m) nominal/actual | variable, see data for actual value |
| Sampling mode / pattern | Static spot ablation |
| Carrier gas | 100% He in the cell, Ar make-up gas combined in cell above ablation in funnel. |
| Cell carrier gas flow (l min$^{-1}$) | 0.5 l min$^{-1}$ |

| ICP-MS Instrument | |
| --- | --- |
| Make, Model and type | Thermo Element XR, Sector-field single collector ICP-MS, with high capacity interface pump |
| Sample introduction | Ablated aerosole direct |
| RF power (W) | 1350 W - 1550W (optimized daily) |
| Make-up gas flow | 0.90 - 1.05 l min$^{-1}$ Ar (optimized daily) 2 ml min$^{-1}$ N$_2$ |
| Detection system | triple (pulse counting, analog, Faraday) cross calibrated daily with U 238, fixed ACF value, all isotopes usually in pulse counting only (<5Mcps) |
| Masses measured (amu) | 202, 204, 206, 207, 208, 232, 235, 238 |
| Integration time per peak/dwell times (ms) | 11 ms (all masses) except: 206, 207 (50ms) |
| Total integration time per output data point (s) | 0.174 s |
| Sensitvity / Efficiency (%, element) | ~ 1 % U |
| Dead time (ns) | 25 |
| Typical oxide rate (ThO/Th) | 0.18% |
| Typical doubly charged rate (Ba$^{++}$/Ba$^+$) | 3.50% |

| Data Processing | |
| --- | --- |
| Gas blank | 20 s |
| Calibration strategy | NIST614 glass standard as primary reference material for drift and Pb-Pb ratios; WC-1 carbonate standard for matrix matching of 206Pb/238U; ASH-15-D and JT carbonate for QC |
| Reference Material info | NIST614 (concentration data Jochum et al. 2011, Pb isotopes Baker et al. 2004); WC-1 (Roberts et al. 2017); ASH-15-D (Nuriel et al. 2020); JT (characterized in this work) |
| Data processing package used | Iolite 2.5, VisualAge for integration, interval selection, and gas blank correction only. In-house spreadsheet data processing. IsoplotR (Vermeesch 2018) for isochrons, intercept ages, and initial Pb compositions |
| Correction for LIEF | No LIEF correction (ratio of mean intensities $^{207}$Pb/$^{206}$Pb, respectively mean of uncorrected ratios $^{238}$U/$^{206}$Pb used) |
| Mass discrimination | normalised to reference material (sample standard backeting) |
| Common-Pb correction, composition and uncertainty | None applied. Ages calculated from regressions in TeraWasserburg concordia plots. |
| Uncertainty level and propagation | Intercept ages are quoted at 2$s$ absolute, propagation is by quadratic addition. Counting statistics uncertainty are propagated to the 207Pb/206Pb ration, together with the uncerainty of the RM value and the uncertainty of repeted measurements. The uncertainty value for lower intercept isochron ages includes uncertainties from the RM, and asystematic uncertainties, estimated in this work to be 2.5%. Decay constant uncertainties are neglected. |
| Quality control / Validation | ASH 15D: mean of $^{206}$Pb/$^{238}$U intercept ages: 2.885 ± 0.076 Ma (2s, MSWD = 0.99, n = 30) (0.9 % Wtd ave uncert. (internal), 2.0% total external uncert.) |
| | JT - mean of $^{206}$Pb/$^{238}$U intercept ages: 13.70 ± 0.37 Ma (2s, MSWD = 1.03, n = 16) (1.15 % Wtd ave uncert. (internal), 2.5% total external uncert.) |
| | Systematic uncertainty for propagation is 2.5% (2$s$). |

This reporting is based on a template available for download on www.Plasmage.org

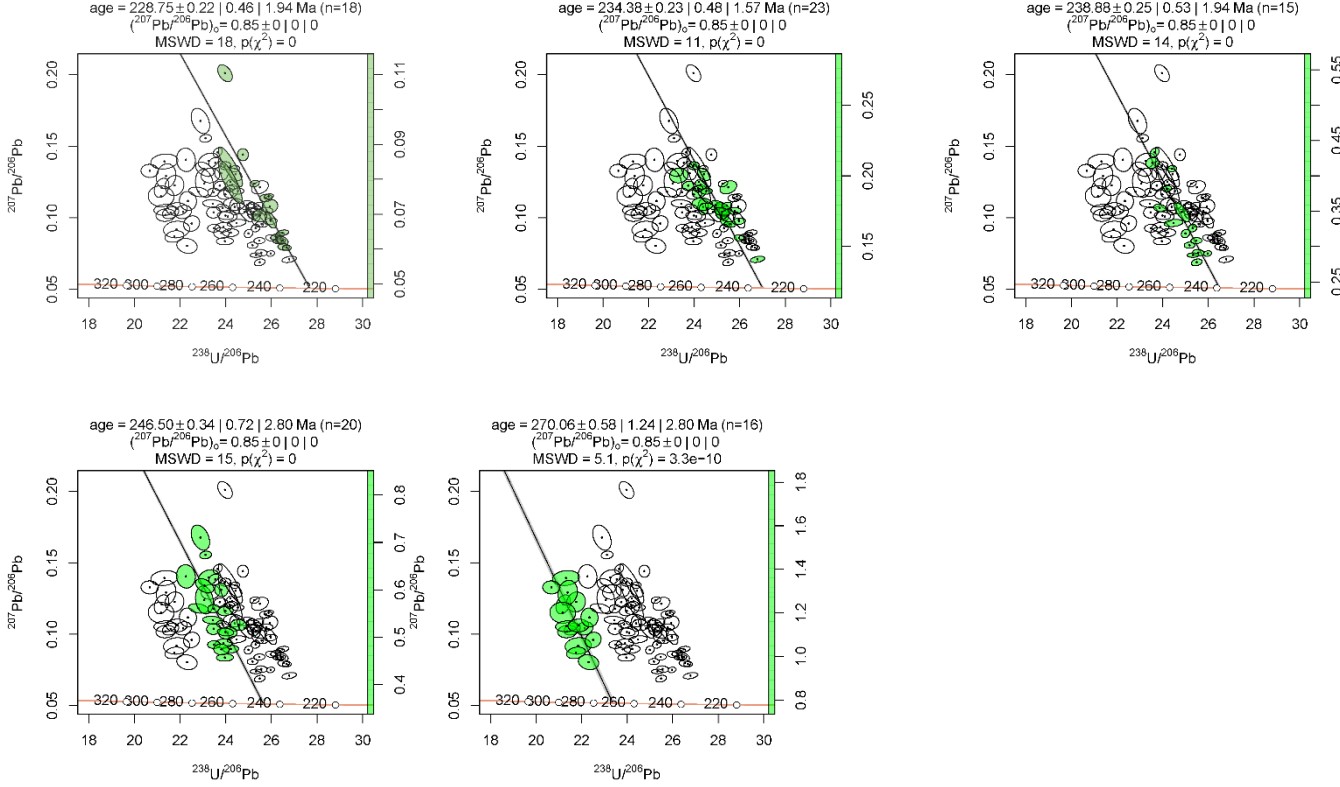

**Figure A1: Intercept statistics of Fig. 2. WC-1 analysed with different aspect ratios.**

## 7 Supplement link

Supplementary material:

Table S1-S2-S3.xls

Additional characterisation of JT.pdf

## 8 Author contribution

Marcel Guillong adapted the LA-ICP-MS methodology and data reduction, designed the LA experiments, carried out most measurements, wrote the manuscript with input from all co-authors. Jörn-Frederik Wotzlaw did the ID-TIMS measurements of JT. Nathan Looser provided the RM JT, prepared the samples, did some LA-ICP-MS and most of the ablation rate measurements. Oscar Laurent did some LA-ICP-MS measurements and was involved in the adaption of the methodology, data reduction process and quality control.

**9 Special issue statement**

**10 Acknowledgement**

Prof. Olivier Bachman and Prof. Stefano M. Bernasconi are acknowledged supporting this work. P. Nuriel is acknowledged for RM ASH-15D, Nick Roberts is acknowledged for RM WC-1. Critical reading and comments by Andrew R Kylander-Clark, David M. Chew and Nick Roberts helped to improve the manuscript.

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
