# Peer review of "Evaluating the reliability of U-Pb LA-ICP-MS carbonate geochronology: matrix issues and a potential calcite validation reference material."

_Geochronology, 2019_

## Referee Comment (RC1) · Andrew R Kylander-Clark (Referee) · 13 Feb 2020

This is a well written, well organized paper with two main points: 1) it describes the down-hole fractionation vs drill rate of various carbonates (primarily calcite), and 2) introduces a new calcite standard: JT. It is relative to a broad community and this is the appropriate platform for publication. I have only a few comments, one which I think is an important point that should be sure to be understood by the calcite geochronologist community. With a few changes, this manuscript should be ready for publication. Comments: 72-73. This is my major comment on the manuscript, and it's possible that I misunderstand the text. It is not entirely clear to me what the authors mean by, "integra-

tion intervals of both reference materials and samples are subsequently adjusted..."
but nevertheless it is critical to point out that because the samples are not corrected
for down-hole fractionation, one cannot adjust the integration interval of the samples or
unknowns unless they adjust it to use exactly the same interval. Because the premise
of this paper is based on the difference in fractionation vs drill rate, it is critical to let
the reader understand that when reducing the data using a glass standard, and using
a secondary correction on the entire 206Pb/238U ratio, it must be assumed that the
same part of the down-hole fractionated analysis is used for both RM and unknowns.
One can imagine a scenario in which the user selects the first half of the analysis for
all reference materials, and the second half of the analysis for all the unknowns. All the
raw dates for the unknown will be younger than the average, because the 206Pb/238U
date gets older down-hole (in most cases - and this is demonstrated rather well in this
paper). All the analyses for the unknowns will be older than the average, and thus the
unknowns will be inaccurate by half of the down-hole fractionation percentage from the
top to the bottom of the hole. Not good. I'm not entirely sure if this is what was done by
the authors or not; their RMs yield accurate ages, possibly because I misunderstand
this line, or because the adjustments were random enough not to make enough of a dif-
ference in the final age. Nevertheless, if this was done the data should be reprocessed
before publication, even though the ages will change imperceptibly (almost surely the
case). 74. "Due to the low 207 count rate" This is a smart way to reduce the data and
a key point in reducing geochronologic data with low count rates. When the data is
particularly noisy, the mean of the ratios yields inaccurate data. This is especially true
with data with a noisy background, as sometimes raw ratios can be negative, which
is physically impossible. Figure 1. The scaling is oh so close, but if you can scale
the two figures so that one can make a direct comparison between the isochrons of
each method, that would be nice. 119. sessions respectively 139. "repetition rate from
110 $\mu$m and 5 Hz for the primary RM WC-1 up to 250 $\mu$m and 5-10 Hz for validation
RM and samples" Please rewrite this for clarity. Did vary the rep rate and spot size
differently for the primary and the unknowns. Just state that variance for each. 143. I

don't know that I'd ever recommend to someone that they run different spot sizes on standards vs. unknowns. Is there a reference here? The demonstration done here is an excellent example of the differences between overall fractionation with different pit depths, but I don't think anyone should recommend that different spot sizes or ablation rates be used between samples and unknowns. 151. But you must not have done the same range in spot sizes and pit depths for the unknowns based on the graph. Please describe a little more clearly the experimental setup. Figure 3 is fine, but Figure 4 should be normalized to the age; i.e., is should be showing the percentage of age offset vs. aspect ratio. Personally, I think it should also go through the origin, that is, one would expect no down-hole fractionation with an infinite width-depth, and the age offset increases from there. This is a more intuitive way of thinking about the process which is being demonstrated. That aside, the a), b) and c) figures are not comparable (slopes), because they are measured vs absolute offset instead of relative offset. Note that the slopes decrease with decreasing age. With relative offsets, one would then be able to note that the slopes are not the same, though they are probably equivalent within uncertainty; it's just that the uncertainty of the latter two datasets in very large. It would be proper to state the slope and its uncertainty and better yet, also add a error envelope on the figure. One can argue that you could draw a horizontal line between the bottom two datasets (though I agree that there must be a slope - it just doesn't match the pit depths in figure 5; both slopes, taken without uncertainties, imply that the pits are deeper in both ASH-15 and JT than they are in WC). 190. I do not endorse this methodology, primarily because of the statement on line 197. Because the ICP parameters and laser parameters can drift, it makes much more sense to use the same spot size for the standards and unknowns. If the unknowns are so low in U such that they need an enormous spot, we should be finding appropriate RMs to deal with that instead of varying the spot size and rep rate. Too many things can go wrong. How many samples are out there that don't work with the currently available RMs? If the authors really want to suggest this methodology, I suggest that they explore the limits of what is possible with RM U, Pb concentrations versus that of unknowns, to show that

there are some samples that cannot be measured with the same laser parameters.

**GChronD**

Interactive
comment

---

## Referee Comment (RC2) · David M. Chew (Referee) · 4 Mar 2020

I really enjoyed reading this manuscript – it is clearly written and laid out and with some very clear figures, and will be a very valuable addition to the U-Pb LA-ICP-MS carbonate spot dating literature. In particular, some of the results on i) the effects of pit aspect ratio on U-Pb age offsets, ii) varying ablation rates for different carbonate matrices and iii) age heterogeneity of WC-1 (which are excellently illustrated in Figure 3, 5 and 7 respectively) are ver significant. I only have some minor comments, as outlined below.

Substantive comments

[Figure]

L72-73 As pointed out by reviewer #1, the adjustment of integrations to optimise spread along an isochron is potentially problematic (depending on how it is done). The downhole fractionation is applied by Iolite to the RM, and then to all unknowns. This could result in age offsets if the downhole fractionation curve for carbonate samples is different to that of the NIST glass primary RM. You could use VizualAge_UcomPbine to see if this is the case, by doing an appropriate 207Pb correction to carbonate samples and seeing if the final 207Pb age channel is consistently 'flat' (i.e. not systematically rising or decreasing)

L81 The drift correction in Iolite is separate and occurs after the downhole correction (see Appendix B1 in Paton et al. 2010, G3)

The slopes in Figure 4 make little sense as presented – use relative (%) age differences on the y-axis

L142-143 This definitely requires a reference or should be deleted. I do not know of any study that uses different spots sizes for unknowns versus RMs which sounds like a pretty bad idea (even if this study subsequently shows that workarounds might be possible as suggested in L189-194). Which bring me on to a question about lines 189-194 – have you tried this?

I would like to see more discussion on what the cause of the excess variance in the two secondary RMs might be. The TIMS data for JT do have quite a high MSWD (incidentally the ASH-15D TIMS data and its MSWD should be briefly mentioned in the text) – is it sampling of a slightly age-heterogeneous material and is this the cause of the LA-ICP-MS excess variance? Also the ASH-15D LA-ICP-MS vs TIMS age offset needs more discussion – the end of the abstract (L19-20) and the end of the conclusions (L241-244) seem somewhat contradictory in this respect.

Typos / minor edits to text

L30 'typically a standard glass' L36 'isochron' L75 do not follow use of the word 'replicate' here L111 define $\varepsilon$' on first usage L119 space between sessions and respectively L124 'inter session' L126 VRM = validation reference material? L149 (and any other occurrences) 'mismatch' L178 'aragonite samples' L216 reword 'usual results'

---

## Short Comment (SC1) · Review of Guillong et al · 9 Mar 2020

Review of Guillong et al

Firstly, this paper presents an important dataset that is of value to the community – a measurement of the potential matrix-effects when measuring U-Pb in carbonate materials. On its own, this would be a useful publication.

Secondly, the paper presents a potential new reference material for U-Pb carbonate geochronology. This is also much needed.

However, a lot of the paper discusses issues with matrix/ablation-matching, and there-

fore the description of the new JT RM is rather scant. How radiogenic is any one region of JT? If JT is distributed, will all aliquots have enough radiogenic lead, and enough spread in U/Pb ratios?

The paper has major flaws in both its presentation, and its message, but which can be very easily amended.

Firstly, using different spot sizes (and hence aspect ratios) for inter-element work, is not common practice within the community and I am not aware of it being advocated as good practice. Early in LA-ICP-MS development (particularly for zircon U-Pb geochronology) papers such as Mank & Mason (1999) and Horn et al. (2000) demonstrated that downhole fractionation affects the resultant U/Pb age, and that drill-rate and aspect ratio are the dominant control on the behaviour of downhole fractionation for any one material. Horn et al. (2000) did in fact demonstrate very broadly that calibration without external normalisation can be achieved. However, this method was soon surpassed (at least by most) by those that advocated matrix and ablation-condition matching (e.g. Jackson et al., 2004). Several other recent papers have tried to tackle ways of reducing inter-element fractionation in zircon U-Pb, e.g. Allen et al. (2012), Solari et al. (2015) and Marillo-Sialer et al. (2014). I do not question the knowledge of the authors on these issues, but the way the issue is presented in this current paper should be addressed to reflect the considerable work that has gone before in controlling and standardising inter-element fractionation during LA-ICP-MS, albeit for U-Pb most of it is through analysis of zircon rather than carbonate.

This paper provides a useful dataset building on the work that has been conducted on zircon regarding matrix-matching, and should be presented in that light. This study documents the scale of the matrix effects (in terms of downhole fractionation) of carbonate materials, and along with the potential new RM, these should be the key messages.

Line 43 – this should be: (e.g. Beaudoin et al., 2018), since many of the listed and not-listed papers (in the previous sentence) state the results from secondary RMs.

(Although technically these are not as yet 'RMs', but merely materials that have been analysed with isotope dilution methods in previous literature).

Line 72 – This line describes a process that has major implications for the accuracy of the data, if, certain precautions are not taken. If a suitable data reduction algorithm is used, that matches the same region of the standard as that of the sample – as in most Iolite U-Pb data reduction schemes, then the accuracy is largely going to be maintained (maybe a minor degradation in accuracy). If the analytical protocol produces a U/Pb profile with no measurable downhole fractionation, then this type of cutting up of the integration can be achieved fairly accurately also. However, the latter is normally only possible with very low aspect ratios or with rasters.

Section 1.4 If this paper is going to be the only paper that presents JT as a RM, then it is rather lacking in detail.

How much JT is available?

How homogeneous is the material chemically and physically?

The data are not dominated by radiogenic lead, but instead provide a spread in U/Pb ratios. This is favourable for generating a precise age, but, will any given 20 spots in one session provide a large spread in U/Pb. i.e. what is the scale of U/Pb heterogeneity across this material?

These factors have to be considered if the material is to be of use to the community. 207Pb counts in JT are presumably quite low. Therefore this material, like ASH15, may have limited use for the Q-ICP-MS community.

Section 2.2 This long-term variance seems reasonable, and about what I would expect based on my own and others general experience.

It is a pretty good validation of the method as a whole, given the overall heterogeneous nature of carbonates.

Line 140-145. "…as sometimes done in carbonate U-Pb dating by LA-ICP-MS". By whom? To my knowledge no one has ever stated this. It is certainly not 'normal' procedure. If it is something that people do, then they need to be more open about this in the description of their method and analytical protocol.

Section 2.3.2

It is a shame more dolomite types were not analysed. This is pending further work I guess, and would ideally involve a dolomite RM. Overall though, these data show some very useful quantification of the potential issues.

Section 2.3.3

"Strategies for matching aspect ratios"

This should just simply state – "match the RMs to the unknowns".

It may not be ideal to match the spot size, due to low U in calcite, but that is just par for the course, and a limitation of the method. Carbonates are a non-ideal U-Pb chronometer. "We suggest a two-step strategy to account for this issue…" This seems like a very time-consuming and problematic workaround:

1) U heterogeneity is often at a smaller scale than the pit depth/diameter. Such that a high U zone in the pre-screening may turn out to be low U for the next 30 seconds of ablation (and vice versa).

2) Chemical and physical heterogeneity can be large, such that the drill rate probably changes a lot during 30 secs of ablation, not just between different materials.

3) This two-step strategy would add a lot of time to the workflow.

4) How is the rep rate adjusted to exactly match the aspect ratio? – for a fixed focus point at the surface of the sample, drill rate is non-linear as you drill down such that exact estimation of depth is difficult.

5) Post ablation measurement of pits will also add significantly more time to the overall data workflow.

6) Applying the correction is not actually tested for some unknowns here, so we don't really know if this two-step strategy is going to be an overall improvement for heterogeneous materials.

7) "A detailed study on how to best apply this correction if necessary is beyond the scope of this work..." – it would appear then that this is basically untested.

The authors critically undermine their arguments in their final sentence of this section: "we suggest for a more robust data reduction to always use similar aspect ratios". I would argue that this is exactly what should have been done, and that the community needs to try and find more RMs with a range of U contents.

Section 2.4

Pb correction – good to see that no offset was observed, this reflects the observations and experience of myself and others but is good documentation of this fact.

Section 2.5 - WC-1 heterogeneity.

The authors state that "This finding demonstrates that WC-1 is not single-phased and heterogeneous in age."

I bring to the attention of the authors the following from Roberts et al. (2017): "A white alteration vein cross‐cutting the mapped region is high in both Th and the transition metals." And "WC‐1 is not the perfect material because of its modest heterogeneity..."

Although this present study does not state that Roberts et al. (2017) claim a single-phased material, one might argue that is implied from their choice of phrasing.

For reference, Roberts et al. (2017) did not use the term 'single-phased', and openly quote the heterogeneity that they observe.

The plots in Roberts et al. (2017) demonstrate that the white Th-rich region analysed in their data is high in common lead, but that the data-points presented seem to be broadly of the right age. It would appear that the author's data shows that some of these altered regions are not the same age. However, rather than a different age being implied, this is just as likely to be variation due to open-system behaviour; an age might not be definable. So, WC-1 may be heterogeneous in age or homogeneous in age with white zones of alteration causing open system U-Pb behaviour in these zones. When this alteration occurred relative to the dated phase is not resolvable and could have occurred quickly after the formation age or sometime after.

I agree entirely that it is worth demonstrating this heterogeneity to others, since WC-1 is now a commonly used RM in the community. However, I would ask the authors to take care that their language better reflects the information presented in Roberts et al. (2017).

As the distributor of WC-1, I apologise for sending out an aliquot that had such a large altered region. I visually screen each piece, but after polishing down, some pieces will have worse regions than others. Altered regions should be avoided, as with any U-Pb or trace element reference material.

Line 241

"offset of 2-3%... of LA-ICP-MS results for ASH-15D compared to ID-TIMS results remains,…"

This sentence is written such that this offset should previously be described. But no description of this is provided. Therefore, what offset? And do the authors have a comment on what this means?

Summary

To reiterate the main points above, what needs changing:

1) The title. As far as I can see, this paper does not present new analytical and data

evaluation protocols. The paper provides data supporting the use of ablation condition matching, which is already the favoured method of most practitioners. Analytical protocols also follow those used by most labs already. The paper discusses a two-step strategy to normalise data with different aspect ratios, but then doesn't actually provide a detailed way of doing this, and goes on to say that not using this method would be better.

The paper does provide: A new potential RM, and data on matrix issues. This should be reflected in the title. e.g. "Evaluating the reliability of U-Pb LA-ICP-MS carbonate geochronology: matrix issues and a potential calcite reference material."

2) Present the data in a different light:

i.e. "we have documented the matrix issues in carbonate dating. . ."

i.e. "we document heterogeneity in the currently used WC-1, and present this information as a caution to analysts to make sure they understand that carbonates are heterogeneous materials, and care needs to be taken. . ."

i.e. "we present analytical data on a potential new calcite reference material. . ."

i.e. "we record a 2-2.5% long-term excess uncertainty, which may provide an ultimate limit on the uncertainty of carbonate U-Pb data. . ."

---

## Author Comment (AC2) · 6 Apr 2020

Thank you for the comments and suggestions that helped to improve the manuscript. Please find below some answers, comments and findings.

Please also see the answers to RC 1, including the figures.

Our method does not use the drift or downhole correction of Iolite. We just use the software to do the baseline subtraction and calculate raw ratios that we subsequently use in an Excel spreadsheet to do the matrix and drift corrections. This is now precised in the text.

[Figure]

"Which bring me on to a question about lines 189-194 – have you tried this? Yes, this is our standard approach to carbonate dating: We load a bunch of interesting samples, program a short sequence with 2-4 points per interesting phase with shortened gas blank / ablation times and run this "test" sequence (20-30 minutes). We monitor live the U and Pb intensities and decide based on the 2-4 point if the tested sample may work, needs larger crater or is hopeless. Criteria for samples worth a shot are $238U>\sim10000$ cps; or $>\sim1000$ cps if there is not much initial Pb, in which case we increase the crater diameter / repetition rate from 110 micron and 5Hz to 163 micron and 7.4 Hz. If $238U < \sim1000$ cps and/or the initial Pb is higher than $238U$, there is little hope, and we skip this sample in the subsequent session.

It is quite common that carbonate samples have a very high MSWD when analysed by ID-TIMS. For instance, the calibration data for WC-1 have a MSWD of 1069 (Roberts et al., 2017). We do explicitly not give the detailed ID-TIMS data for ASH-15 in this manuscript as these data will be published elsewhere (P. Nuriel et al., in preparation)

Thank you for suggesting using the VizualAge_UcomPbine DRS in Iolite, we will give it a try soon.

We adjusted the manuscript with your suggestions. VRM is defined on Line 10 in the abstract.

---

## Author Response (AR1)

Answers to RC1:

72-73: We completely agree that downhole fractionation is an important effect to be considered in U-Pb dating, which has been shown in many details in the case of zircon, and that it is necessary to be extremely careful with its correction to get the best possible accuracy. However, we observed in the case of carbonate U-Pb dating some differences:

1. Downhole fractionation of NIST 614 and any carbonate material is not comparable, therefore, any attempt to correct a carbonate LA signals using NIST 614 can introduce systematic errors. Even the downhole fractionation trend between different carbonates can be different Fig. RA1

2. Due to the variability of different carbonate minerals and textures, the ablation rate can vary as is shown in the manuscript. Therefore, an attempt to match integration intervals between standard and samples will unfortunately not be perfect as it might be different with respect to downhole fractionation.

3. We agree that if all carbonates showed the same downhole fractionation pattern the approach to get best accuracy is to use exactly the same interval, but due to heterogeneity in ablation rate, within-run variations in initial Pb, U content, and age variations this is hardly possible: a comparison for one piece of JT (10 ablations) is presented comparing total integration with selective and multiple integrations. Fig. RA2

4. If there are high- and low initial Pb zones within a single ablation (which is not uncommon), the total interval will show a large uncertainty with little contribution to a well defined isochron. Two separate integration intervals for high initial Pb and low initial Pb will result in 2 measurement points contributing to a better-defined Isochron, even if the matrix/downhole correction with the total interval from the reference material will add a systematic offset. However, we assume that the variability of initial Pb is randomly distributed from a crater to another and therefore the possible offset is averaged out. Fig. RA2. This might become a problem when the "initial" Pb is mainly at the beginning of the signal due to surface contamination. All precaution should be taken to minimize surface contamination (careful cleaning and pre-ablation).

5. One possible solution would be that only signals where there is minimal variation within the ablation for initial Pb are considered, but then quite a lot of ablations will have to be discarded.

6. We agree on the referee's point that the "adjustments were random enough not to make enough of a difference in the final age". We also agree that it can be possible by selecting only later parts of the signal, the final age will get older respectively younger when only the first parts of the signal are selected. However, the new data of JT, for which we systematically investigated this effect, does not show this behaviour, although data on e.g WC-1 with a clear increasing downhole fractionation trend (Fig RA1) will certainly show this effect. Fig. RA3

7. Figure RA4: shows the downhole trend of WC-1 for different aspect ratios (mean of n>15 signals) both comparing vs time (A) and vs the estimated aspect ratio (B). This figure shows that up to an aspect ratio of 0.2-0.3 no clear downhole trend is observed and only for higher aspect ratios the Pb/U increases significantly.

Therefore, we changed the description accordingly: "The selection of different integration intervals along a single hole ablation can introduce systematic offsets if not randomly distributed due to different amounts of downhole fractionation between RM and sample if there is significant amount of downhole fractionation in either the RM and/or the sample. Best practice is to use as good as possible the same integration intervals with respect to crater shape for both

the RM and the sample. As is demonstrated, it is likely that random variability of downhole fractionation, ablation rate, distribution of initial Pb etc. would anyway mitigate the offset potentially introduced. This potentially introduced offset would anyway be diluted in the propagation of the systematic uncertainties, especially since the long term excess variance of secondary RM could precisely result from this.

We improved the manuscript according to some detailed comments.

143: The rather controversially comment "as sometimes done in carbonate U-Pb dating by LA-ICP-MS" has its origin in both personal communications and some unfortunately rather vague descriptions of analytical methods in some publications (like e.g. Nuriel et al., 2017: "either 85 or 110 micron spot size" but no clear statement about whether this is for different sessions or a single one; https://doi.org/10.1130/G38903.1). Therefore, it is of course clearly not suggested in literature to use different crater sizes until now, but this may not prevent users to do so, so we felt like this was important to stress out. To temper this, we adjusted the statement, and can recommend to use different spot sizes and repetition rate as long as the aspect ratio is the same as the reference material.

Figure 4: We adjusted the Y-axis to offset from the ref. value. The associated uncertainties for both the age and the aspect ratio mismatch are large. Especially, the uncertainty of the aspect ratio mismatch relative to the RM is difficult to quantify. Indeed, these values result from a combination of estimations based on the number of pulses and an average ablation rate for this sample, itself based on measurements made for some (but not all) craters in some (but not all) sessions. We think that measuring the depth of each individual crater as precisely as possible would be a considerable amount of work considering the small influence that it would have on the final results. Therefore, we do not give an uncertainty on the x value and also refrain from calculating an uncertainty on the slope. Based on the new slopes, we think they are equal within uncertainty.
Figure 4 is also meant to show the general effect observed and described in this manuscript that with changes of the crater geometry you likely get significant age offsets. However, the ablation rate of the carbonates is extremely variable (as shown in Figure 5) and can vary both from spot to spot of the sample sample, as we occasionally observed. The figure qualitatively shows that the described effect is present, and that with different aspect ratios an offset is introduced.

What the referee 1 suggests is in the end a perfect matrix matching between RM and sample, which considering all the sources of uncertainty above, is in our opinion not possible. What we can suggest is therefore a reasonable approach to get as accurate results as possible for many interesting applications with the presented approach.

To be able to adjust the crater size is an advantage that makes LA-ICP-MS the method of choice for many applications that we would not sacrifice this versatility, but obviously care has to be taken. If the grain size of a sample is highly variable we like to apply different crater sizes to get trace element content for low concertation with larger spot size and good reproducibility for higher trace elements with many replicates of small craters. In the end there have to be some compromises and we think that in the manuscript we show a way to use different sample sizes without sacrificing the accuracy too much by adjusting as good as possible the crater geometry. We think the dependence of the uncertainty on the signal intensity is well shown in literature and with larger craters a higher ablation rate more sensitivity gives smaller uncertainty.

In the end, we are confident that using the excess uncertainty as recommended in this manuscript would anyway cover the uncertainty associated with the differences between RM and most samples, including differences in downhole fractionation, ablation rate, integration intervals, etc. – as long as the aspect ratio of the craters is kept similar and the whole analytical protocol is followed as good as possible.

[Figure]

Figure RA1: Relative to the mean raw 206/238 ratio) downhole fractionation trend for 3 different Carbonates and NIST 614 showing no consistent trend, making a downhole fractionation correction of unknown samples very difficult.

[Figure]

Figure RA2: Comparison of 10 JT ablation once integrated the whole signal (A) and once integrated to get max. spread of Pb/U ratios including multiple integration intervals per signal showing a clear improvement in precision. Colour code is the same ablation have the same colour.

[Figure]

Figure RA3: Individual, isochronal initial Pb corrected JT ages vs. the mean time of integration after laser starts ablating. Blue points indicate selective integrations often the first part (around 8 seconds) and the second half of the signal (around 24 seconds). No significant older ages were found for integrations in the second part. Red points indicate the same for always integrating the whole signal.

[Figure]

Figure RA4: The downhole fractionation trend of WC-1 for different aspect ratios ( mean of n>15 signals, different crater sizes and repetition rate from 163 microns at 3 Hz to 51 microns and 15Hz) both comparing vs time (A) and vs the estimated aspect ratio (B). This figure shows that up to an aspect ratio of 0.2-0.3 no clear downhole trend is observed and only for higher aspect ratios the Pb/U increases significantly.

Answers to RC2:

Please also see also answers to RC 1:

Our method does not use the drift or downhole correction of Iolite. We just use the software to do the baseline subtraction and calculate raw ratios that we subsequently use in an Excel spreadsheet to do the matrix and drift corrections. This is now precised in the text.

"Which bring me on to a question about lines 189-194 – have you tried this?
Yes, this is our standard approach to carbonate dating: We load a bunch of interesting samples, program a short sequence with 2-4 points per interesting phase with shortened gas blank / ablation times and run this "test" sequence (20-30 minutes). We monitor live the U and Pb intensities and decide based on the 2-4 point if the tested sample may work, needs larger crater or is hopeless. Criteria for samples worth a shot are $^{238}$U>~10000 cps; or >~1000 cps if there is not much initial Pb, in which case we increase the crater diameter / repetition rate from 110 micron and 5Hz to 163 micron and 7.4 Hz. If $^{238}$U < ~1000 cps and/or the initial Pb is higher than $^{238}$U, there is little hope, and we skip this sample in the subsequent session.

It is quite common that carbonate samples have a very high MSWD when analysed by ID-TIMS. For instance, the calibration data for WC-1 have a MSWD of 1069 (Roberts et al., 2017). We do explicitly not give the detailed ID-TIMS data for ASH-15 in this manuscript as these data will be published elsewhere (P. Nuriel et al., in preparation)

Thank you for suggesting using the VizualAge_UcomPbine DRS in Iolite, we will give it a try soon.

We adjusted the manuscript with your suggestions. VRM is defined on Line 10 in the abstract.

Answers to SC 1:

Thank you for the comments and suggestions that helped to improve the manuscript. Please find below some answers, comments and findings.

Please also see also answers to RC 1 and 2:

Thank you for pointing out parts of the long and intensively discussed, investigated literature on inter element fractionation. We are well aware of the large amount of interesting and good literature about it, but we would not like to write a review manuscript on this topic so we kept the referencing and discussion to the necessary part, also with respect to the length of the manuscript. Additionally, we investigated downhole fractionation in carbonates, but no clear results emerged (e.g. Figure RA 1) or some investigations using the matrix matched synthetic reference material of pressed powder MACS-3 which is "completely useless for U/Pb dating of carbonates" due to heterogeneity and non-reproducible U-Pb fractionation behaviour.

We added some more detailed investigations on the JT samples and pieces available for distribution, including an image of all available pieces and results of 10 analyses on all possible aliquots that can be distributed. Part of our work over the past couple of years consisted in investigating the suitability of other possible secondary RM, but so far all tested materials but JT were unsuitable.

We added a whole new chapter to the Electronic appendix describing the available pieces of JT in more detail. We agree that JT is both of limited supply and use for low sensitivity ICP-MS, however we think that it is with all the limitations a valuable addition to the collection of possible RM and usable as VRM for some laboratories.

Line 72: We now state:

"The selection of different integration intervals along a single hole ablation can introduce systematic offsets if not randomly distributed due to different amounts of downhole fractionation between RM and sample if there is significant amount of downhole fractionation in either the RM and/or the sample. Best practice is to use as good as possible the same integration intervals with respect to crater shape for both the RM and the sample. As is demonstrated, it is likely that random variability of downhole fractionation, ablation rate, distribution of initial Pb etc. would anyway mitigate the offset potentially introduced. This potentially introduced offset would anyway be diluted in the propagation of the systematic uncertainties, especially since the long term excess variance of secondary RM could precisely result from this."

Line 140-145. See answer to RC1 about this topic.

*Section 2.33*

1) *U heterogeneity is often at a smaller scale than the pit depth/diameter. Such that a high U zone in the pre-screening may turn out to be low U for the next 30 seconds of ablation (and vice versa).*

We agree, but the opposite is also true: a low U signal during screening may have yielded a higher U signal if ablated longer. This is why we systematically ablate several screening spots per sample, to get an idea of not only the U and initial Pb contents, but also their variability. This fast pre-screening gives some first hints and indications of the possibility if the sample is datable or not, which saves a lot of spot programming and analytical time. From personal communications, we know that other laboratories apply similar strategies.

2) *Chemical and physical heterogeneity can be large, such that the drill rate probably changes a lot during 30 secs of ablation, not just between different materials.*

Yes, we agree, as discussed earlier for replies to other referees. We believe that this is at least partly the reason for the larger excess uncertainties we obtain here for carbonate U-Pb dating, compared to zircon geochronology.

3) *This two-step strategy would add a lot of time to the workflow.*
4) The extra time that this short test sequence costs is much less compared to the time lost if samples with very low U are present and not identified. So we would argue that on the long run, this two-step strategy saves time and resources. *How is the rep rate adjusted to exactly match the aspect ratio? – for a fixed focus point at the surface of the sample, drill rate is non-linear as you drill down such that exact estimation of depth is difficult.*

The rep rate is adjusted assuming equal ablation rate per pulse independent of the crater size and depth. Ablation rate variations due to laser focus are generally small in LA up to an aspect ratio of 1 (i.e. for most aspect ratios considered in this study and carbonate dating in general, for which large spot diameters are used), and likely more influenced by the sample properties (Horn, 2001).

5) *Post ablation measurement of pits will also add significantly more time to the overall data workflow.*

Yes, we do not suggest doing this routinely and this would not be necessary anyway if one follows our suggestion to match aspect ratios between RM and unknowns. The depth measurement was specifically performed here to evaluate the importance of the ablation rate and aspect ratios.

6) *Applying the correction is not actually tested for some unknowns here, so we don't really know if this two-step strategy is going to be an overall improvement for heterogeneous materials.*

We assume that this comment is about the correction for similar aspect ratio, and if so, yes we do not show data for "heterogeneous" unknowns and only for Ash-15D and JT. Of course, we cannot exclude that strong heterogeneity in ablation rates for unknowns would result in age offsets. However, as discussed above (reply to reviewer 1) this effect is likely to be mitigated between different spots and matching as good as possible the aspect ratio of the unknowns to the RM is the best way to minimize it.

7) *"A detailed study on how to best apply this correction if necessary is beyond the scope of this work. . ." – it would appear then that this is basically untested. The authors critically undermine*

*their arguments in their final sentence of this section: "we suggest for a more robust data reduction to always use similar aspect ratios". I would argue that this is exactly what should have been done, and that the community needs to try and find more RMs with a range of U contents.*

What is beyond the scope of this manuscript is to present a method that includes a correction based on crater depth measurements post ablation. What is improving the data quality and versatility is to match the aspect ratio of the pre tested samples to the RM. Using this the amount of ablated material and intensity of U can be adjusted closer to that of the RM. As long as there are not more RM with matching U content and ablation behaviour are available this is in our opinion the best possibility to get best possible precision and accuracy. Additionally, the huge variability of U contents in carbonates, and the fact that we don't really know how high it is a priori is a good argument to for our approach. Matching aspect ratios, rather than spot diameters and U content with a series of RM, is much easier and more cost-/time-efficient at present and is the only option having only one reliable primary RM (WC-1) in the community.

We rephrased to make this point clearer.

Section 2.5:

We rephrased this section to better reflect the information presented in Roberts et al. (2017) concerning the white zone, added the isochrone to the white part and the ages for comparison.

"The plots in Roberts et al. (2017) demonstrate that the white Th-rich region analysed in their data is high in common lead, but that the data-points presented seem to be broadly of the right age. It would appear that the author's data shows that some of these altered regions are not the same age. However, rather than a different age being implied, this is just as likely to be variation due to open-system behaviour; an age might not be definable. So, WC-1 may be heterogeneous in age or homogeneous in age with white zones of alteration causing open system U-Pb behaviour in these zones. When this alteration occurred relative to the dated phase is not resolvable and could have occurred quickly after the formation age or sometime after."

We are convinced that the two phases (the dark and the white zones) show carbonate precipitation at two significantly different points in time / in the burial history of the Capitanian Reef Complex. The black zone (which is the regular part of WC-1 that should be used as RM) is composed of marine botryoidal cements (Roberts et al. 2017) while the white zone is composed of vein-like, more sparry cements (again showing two zonings of different luminosity as can be seen by cathodoluminescence microscopy) with sharp contacts to the surrounding botryoidal cements. If the difference in U-Pb between the botryoidal cements and the more sparry white cements would just be the result of open system behavior, the two cements would not show the difference in texture reflecting different ambient conditions during precipitation that we observe (e.g. crystal size, shape). Also, the U-Pb data of the younger (leached) phase would not define an isochron (which it more or less does, aga 203 +/- 7 Ma) but rather be random distributed towards younger ages. The larger scatter in the data of the white zone can potentially be explained as artefact due to mixture of the dark and white phases in the lower part of the ablation pits (the vein might be tilted) and by the fact that later-diagenetic phases like veins in our experience are commonly more noisy compared to WC-1. We therefore argue that WC-1 is heterogeneous in age. However, the white zones can easily be seen by cathodoluminescence (it has a very bright luminosity compared to the botryoidal cements) so that it is not a problem for the community to avoid the white zones.

Line 241: It is the offset of the average LA-ICP-MS age for ASH15-D compared to the ID-TIMS age. As the ID-TIMS results for ASH15 are not part of this manuscript but in (Nuriel et al., in prep.) and there is an offset observable in both Figure 2b and 4c, we do not ignore this but mention that there is this offset, and a possible explanation is given in the conclusion: "This offset cannot be explained completely by differences in ablation rate and may be an additional matrix effect to be investigated in detail in future work." This offset is already mentioned in the Abstract: "Additionally, a systematic offset to the ID-TIMS age of 2-3% was observed for ASH-15D but not for JT. This offset might be due to different ablation rates of ASH-15D compared to the primary RM or remaining matrix effects, even when chosen aspect ratios are similar."

We consider changing the title as suggested and present the data in a different light.

Attached is a file and table S3 that we would like to add to the electronic supplementary information of the main manuscript containing analyses of all the available parts of JT.

Answer to Editor:

Dear Axel,

Thank you for the additional comments, which I answer below in detail.

"However, two invited experts plus an additional comment have pointed out various points where there is some ambiguity, details need a better explanation or have not been correctly presented. Besides, they have made several suggestions for improvement or clarification. In their reply, the authors have clarified most but not all of these issues mentioning they have fixed this in a revised version. This discussion is in many parts useful for the reader, and I would encourage the authors to implement it as much as possible in the manuscript (or external data depository)."

We implemented as many comments and parts of the answer to the referees as we think is good for the manuscript without going into too much speculation about downhole fractionation. Especially we included the additional measurements on JT on the actual pieces that are available for distribution.

"I agree with most comments/explanations of the authors in discussion with the reviewers and as well with most statements made in the manuscript, but I recommend using careful language and clarifying critical issues, e.g. those express by Nick Roberts."

We think that with respect to the initial characterisation of WC-1 by Nick Roberts we now have clarified a some points so that it is clear that he mentioned "WC-1 is not the perfect material because of its modest heterogeneity". As we see 238/206 ratios ranging from 19 to 34 (almost 50%, compared to the published 25%) and the intercept ages of 203 instead of 254 Ma is about 20% off. We think this is not only a "modest" heterogeneity and likely not only an open system behaviour as the points clearly do not fall on a single mixing line, but rather two. For this we improved Figure 7 now showing two Isochrones.

The text on the WC-1 Heterogeneity is improved:
"We document heterogeneity in the currently most used calcite RM WC-1(Roberts et al., 2017). This information is meant as a caution to analysts to make sure they understand that carbonates are quite heterogeneous materials and are prone to diagenetic alteration including open-system processes at various stages in their geological history resulting in zones with the potential for different ages being

recorded. Care needs to be taken as some WC-1 aliquots may have larger age variation than initially described (Roberts et al., 2017)."

"Similar as the reviewers, I am skeptical about using different ablation conditions between RM and unknowns. The lab in Frankfurt, as an example, has shown that it is possible routinely to use the same spot size, energy density, and frequency between RM and unknowns (perhaps scarifying the spatial resolution in some cases) despite of low or high U concentration. The concept of aspect ratios, as applied in this study, is not common practice at all, and there is no fundamental detail study that has validated it (if I am wrong, please cite it)."

While we understand the skepsis to the presented method and we respect the excellent work done in Frankfurt we would like to point out that it is not only the spatial resolution in some cases that is sacrificed, but the possibility to change the amount of introduced material. There is a well known correlation between sensitivity and precision in ICP-MS. The proposed method, if used carefully and similar aspect ratios are used can be used to date samples that were previously not possible ore only possible with inferior precision using the identical crater size either due to limitations from low count rates or detector cross-calibration issues (practical usable linear dynamic range of the detector). The main focus of the manuscript is not to promote the use of different crater sizes but the observation and warning that in the rare case when using different crater sizes (for whatever reason) without matching the aspect ratio an offset is introduced, so if it is necessary to change the crater size the aspect ratio should be match by changing the repetition rate to improve accuracy.

"The presented data is not entirely convincing; applied to ASH 15D with WC-1 as RM, the ages are 3-5% too young, and in case of JT, ages do not change in the aspect ratio miss-match of 0.8 to 1.7 (due to higher uncertainty, but still its no validation…)."

While we agree that we have an offset for ASH-15D we think that overall, the correlation between aspect ratio offset and age mismatch is quite convincing, even if the uncertainties for ASH and JT are overlapping within the investigated range, the trend is clear for these 2 samples and definitively convincing for WC-1.

Secondly, the authors claim due to different drill rates observed for dolomite (up to 1.6 aspect ratio miss match) and aragonite (1.9, n = 1!) a significant age offsets of 4-8% for dolomite and even higher for aragonite. However, they have not demonstrated it! Besides, the authors not demonstrated that different drill rates using a fixed spot diameter (e.g., for WC-1) yielded the same age offset as by varying both spot diameter and drill rate (see Figure 4). So they link two different observations, which cannot simply be linked without validation. I would be grateful if the authors could add something to the text to make their case stronger. However, I am already happy when they express it in a more careful language and mentioning that further validation is needed.

We agree that we so far did not validate any age offset for dolomite and aragonite due to differences in ablation rate when using identical ablation parameters. Therefore we now use a more careful language only suggesting this effect, pointing out that this needs validation.
"We presume that these differences in ablation rates could result in significant age offsets in a roughly estimated range of 4-8 % for dolomite (160 % ablation rate compared to WC-1)and 6-11 % for aragonite (200 % ablation rate of WC-1) based on the offsets found in figure 4. However, this hypothesis needs validation and any attempt to date dolomite or aragonite needs careful validation."

Please see the revised manuscript with all changes we propose, including a more suitable title:

"Evaluating the reliability of U-Pb LA-ICP-MS carbonate geochronology: matrix issues and a potential calcite validation reference material."